# Apparatus Design of One-Step Double-Side Friction Stir Welding for Aluminum Plates

Nurul Muhayat [1], Ericha Dwi Wahyu Syah Putri [1], Hendrato [2], Yohanes Pringeten Dilianto Sembiring Depari [2], Poppy Puspitasari [3], Jamasri [4], Aditya Rio Prabowo [1] and Triyono [1],*

1    Mechanical Engineering Department, Universitas Sebelas Maret, Surakarta 57126, Indonesia; nurulmuhayat@staff.uns.ac.id (N.M.)
2    National Research and Innovation Agency of Republic Indonesia (BRIN), Jakarta Pusat 10340, Indonesia
3    Department of Mechanical and Industrial Engineering, Universitas Negeri Malang, Malang 65145, Indonesia
4    Department of Mechanical and Industrial Engineering, Universitas Gadjah Mada, Yogyakarta 55281, Indonesia
*    Correspondence: triyono74@staff.uns.ac.id

**Abstract:** Aluminum alloys emerged as one of the materials used in manufacturing automotive car bodies due to their advantageous properties such as high strength-to-weight ratio, relatively low cost, high ductility, and high corrosion resistance. However, joining aluminum alloys using fusion welding poses serious problems due to the high solubility of hydrogen gas, which causes porosity in welding metal. Subsequently, solid-state welding, such as friction stir welding (FSW), has been considered a porosity-free aluminum joining method. However, the method has limitations, such as low flexibility and the need for a complex clamping system. It is particularly problematic when welding plates. It causes the welding process to be carried out twice on opposite sides, resulting in longer production times. This study designed and assembled a one-step double-side FSW apparatus to address this challenge and conducted welding trials with various welding parameters. During the welding trial, the upper and lower tool rotation varied at 900/900 rpm and 1500/1500 rpm. As a result, one-step double-side FSW was successfully used for welding 6 mm aluminum without any porosity defects. Faster tool rotation results in a wider heat-affected area and higher tensile strength. In addition, the hard test showed that the one-step double-side FSW process had a lower hardness compared to the hardness of the base metal.

**Keywords:** design; one-step double-side friction stir welding (FSW) machine; lightweight vehicle car body; aluminum; porosity

## 1. Introduction

Using lightweight materials such as aluminum for vehicle car bodies can significantly reduce fuel consumption, resulting in faster and more efficient transportation [1–3]. Therefore, vehicle car bodies for mass passengers such as buses and high-speed trains (HST) are made of aluminum. Typically, welding is the preferred method for joining the various components of these car bodies. In aluminum welding, the main challenge is the formation of porosity caused by the very high solubility of hydrogen in the molten aluminum and the hydrogen gas trapped in welding metal when it solidifies and forms pores [4–12]. An example of porosity defects in aluminum fusion welding is shown in Figure 1. It is important to observe proper welding procedures to avoid porosity in aluminum welding. Ensuring good surface preparation, properly controlling weld parameters, using an effective shielding gas, and choosing the right filler material are some of the steps that can help reduce the likelihood of porosity occurring.

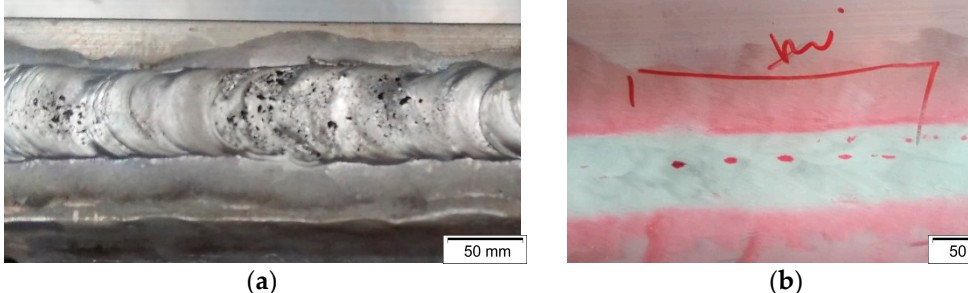

**Figure 1.** Porosities: (**a**) surface porosity, (**b**) internal porosity appeared after penetrant test [13].

Porosity defects in aluminum welding can be overcome by semi-solid welding methods such as friction stir welding (FSW), which was discovered by The Welding Institute (TWI) in 1991 [14]. The working principle of FSW is to generate thermal energy by applying the friction of the rotating tool to the workpiece surface until it becomes soft (Figure 2a). When the joined material is in soft conditions, a pin at the tool's end is used to stir the material. It causes the material to flow at the interface of the two joined materials and form a joining. Since the materials are in semi-solid conditions, hydrogen gas cannot dissolve into welding metal, so porosity defect is not included [15,16]. However, despite its benefits, FSW has some limitations, such as low flexibility and high clamping complexity, and can only be used for welding in a flat position. Moreover, welding in vertical and overhead positions is limited, and FSW can only be performed on one surface of the joined plate. When applied to a thick plate, the pin length becomes a limiting factor, and the specimen must be reversed to perform on both sides of the plate. Reversing a large welded structure, such as a car body for a bus or train (i.e., underframe, side wall, and roof), requires significant energy, time, and production risks [17,18]. It is important to design an FSW mechanism that can work on one step on two sides simultaneously in one welding run to solve this problem, known as a one-step double-side FSW, as seen in Figure 2b.

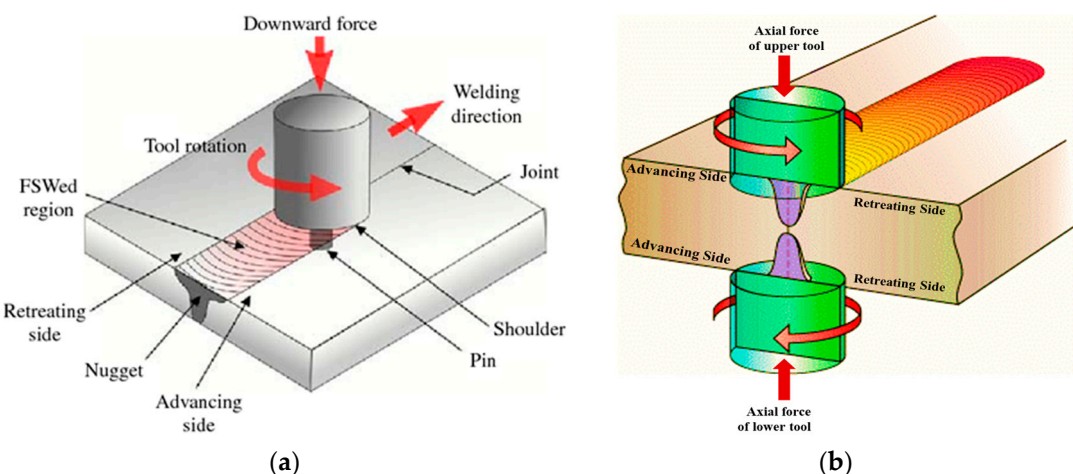

**Figure 2.** Friction stir welding: (**a**) existing method, (**b**) proposed method [19].

The previous studies aimed to increase the flexibility of the FSW process and computer simulations for the motion mechanism of the FSW tool have proven to be very helpful in developing this method [20]. Improving the FSW process is carried out by reducing the resulting defects. These defects arise as a result of the selection of parameters that are not suitable. The parameters that affect FSW welding results are rotational speed, welding speed, tool deep plunge, tilt angle, and tool geometry [21]. Prabha et al., 2018 researched the effect of tool rotational speed on the mechanical properties of AA 5083 weldments in friction stir welding [22]. The lower rotational speed results in lower heat input and defects such as cracks and pinholes in the friction stir processed zone and resulted from lower

tensile values. In addition to the rotational speed of the tools, the tool geometry affects the performance of the similar and dissimilar friction stir welding process to improve the microstructure and mechanical properties of the joints. Therefore, the complex material flow behavior at different welding levels of different aluminum alloys will affect the mechanical properties of the welded joints [23]. Figure 3 shows the development of the FSW research process over a decade to increase the welding result.

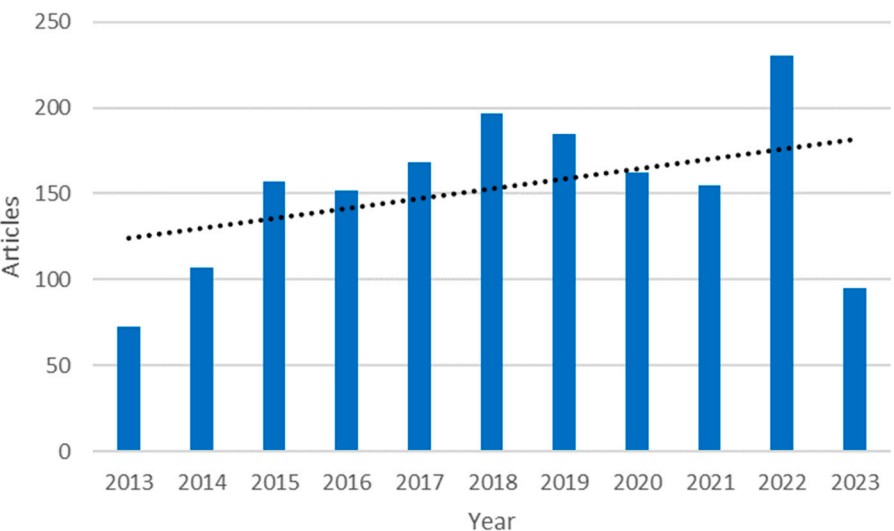

**Figure 3.** Progress research of friction stir welding (2013–2023).

However, most of the existing FSW research has been conducted on thin plates despite the fact that the field applied thick plates such as cryogenic fuel tanks, airframes, nose cap shells, and composite-wrapped high-pressure vessels [24]. To increase its strength, a double-side FSW process is required. Hejazi et al. (2016) investigated that the tensile strength of two-side FSW joints has a strength of 41% greater than that obtained for the single-side FSW joint [25]. This method can be carried out using two-sided FSW joints, namely the bobbin design tool or the conventional FSW method with a two-step welding process. A bobbin-type tool is used, so there is no need for a backing plate, but it is hard to use this tool under the condition that there is limited access to the back of the weldment. Although the conventional FSW method with a two-step welding process is flipped over and twice processed on both sides of the plate, this process takes a large amount of production time and energy, so it is necessary to develop a one-step double-side FSW method. In this method, two tools move simultaneously on the two sides of the plate being welded so that the joining process can be performed with one welding step.

In contrast, the movement mechanism of a one-step double-side FSW tool is more complex as the tool must simultaneously friction, stir, and move along both sides of the joined plate to form a butt joint. Although limited information is available on the apparatus and mechanism of the double-side FSW method, this study will provide an illustration of its design and mechanism, evaluate the trials of influential parameters, and optimize the process. During the welding trial, aluminum alloy AA6061 was used, and the upper and lower tool rotation varied at 900/900 rpm and 1500/1500 rpm. The joints were evaluated by using radiographic testing, macro and microstructure evaluation, and mechanical testing.

## 2. Materials and Methods

### 2.1. Apparatus Design and Assembling

The design was carried out in a sequence, according to design standards of machine elements, including the identification of problems, the selection of parts movement mechanisms, the calculation of forces and stresses that occur in parts, the selection of materials, the measure of the parts of dimensions, as well as finally fittings and assembling of parts.

Based on the tool motion mechanism of one-step double-side FSW as seen in Figure 2b, the apparatus and the movement mechanism of its parts were designed using SolidWorks software. The geometry, size, and shape of the main parts of the apparatus were determined with this software. Tool movement mechanism, workpiece table, and support plate were modeled to evaluate the efficiency of the motion path. Furthermore, the design of the power machine was performed on the main parts of the apparatus. The design is based on trial data from a single-side FSW machine, where to achieve a good FSW welding joint, the pressure on the tool post must be set at 2.5 MPa. By applying a safety factor of 2, the operating pressure of the machine is 5 MPa. The friction coefficient between the tool surface and the workpiece is assumed to be 0.3. In order to obtain the required motor power, the following calculations are performed: calculating the compressive force of the tool on the workpiece using Equation (1), determining the friction force between the workpiece surface and the tool using Equation (2), calculating the maximum torque on the tool using Equation (3), and, finally, calculating the minimum motor power using Equation (4).

The selection of the material was carried out based on the existing single-side friction stir welding. The main components of the machine body used AISI 1030 steel because it has high strength and hardness, low price, and good machinability and weldability. AISI 1030 steel has a yield strength of 440 MPa, which is significantly higher than the stresses that occur during welding. The selection of this material aims to ensure that the machine is safe, rigid, and has no vibrations. After the selection of the material, purchasing materials for the apparatus parts was then performed. The raw materials were then machined to create designed shapes and sizes, and some parts were purchased in semi-assembling condition. Finally, the parts were assembled into a machine. The flow chart of the machine manufacture is shown in Figure 4.

$$F = P_t \times A \tag{1}$$

$$F_f = F \times \mu \tag{2}$$

$$T = F_f \times r \tag{3}$$

$$P_m = \frac{2\pi n T}{60} \tag{4}$$

where F is the compressive force of the tool to the surface of the workpiece, $P_t$ is the pressure of the tool, A is the cross-section area of the tool, $F_f$ is the friction force of the tool on the workpiece surface, $\mu$ is friction coefficient between the tool and workpiece surface, T is torsion, r is the radius of the tool, n is the tool rotational speed, and $P_m$ is the motor power.

*2.2. Welding Trials and Parameter Process Evaluation*

The assembled one-step double-side FSW machine was tested with potentially influential welding parameters on both the first and second tools, including tool rotation speed, transverse tool speed, and tool plunge depth. The solution for the test was carried out by varying the rotation speed of the upper and lower tool. In contrast, other parameters, such as transversal speed, tilt angle, and plunge depth, were kept constant at 30 mm/min, 2°, and 2.0 mm, respectively. The rotation speed of the upper and lower tool varied from 900/900 and 1500/1500 rpm in the same direction.

Aluminum alloy 6061 with a thickness of 6 mm was selected as the study material. This material was chosen because it has good strength, formability, corrosion resistance, weldability, and easy recycling. Therefore, AA6061 is widely applied to vehicle structures, especially car-body trains. The chemical composition of this material is shown in Table 1 [25]. The FSW tool was cylindrical straight and made from AISI H13 steel with increased hardness and wear through heat treatment. The shoulder diameter, pin diameter, and pin height were 20 mm, 5 mm, and 2 mm, respectively. The detailed dimension of the tool is shown in Figure 5.

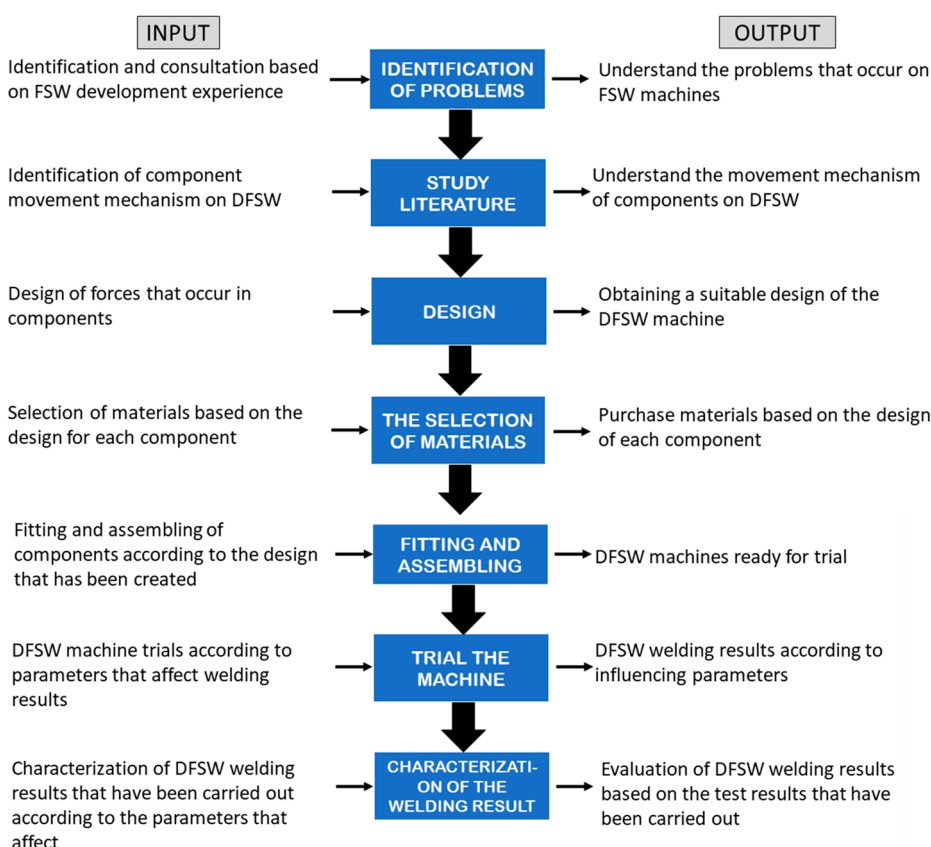

**Figure 4.** The flow chart of the one-step double-side FSW machine design.

**Table 1.** Chemical composition of AA6061-T6 (wt%) [26].

| Element | Al | Si | Cu | Fe | Mn | Mg | Ti | Zn |
|---|---|---|---|---|---|---|---|---|
| wt% | 97.19 | 0.63 | 0.24 | 0.52 | 0.08 | 0.94 | 0.06 | 0.04 |

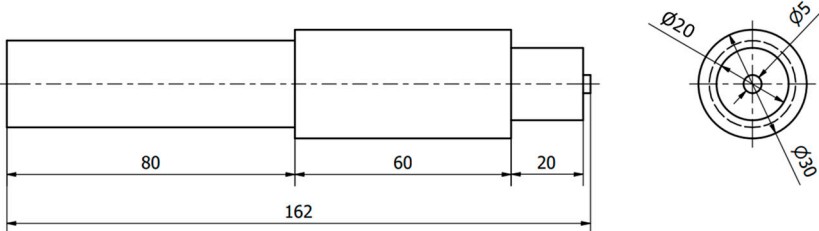

**Figure 5.** Detail dimension of the double-side FSW tool.

The welding coupon test has a dimension of 100 mm × 200 mm × 6 mm with a butt joint, as seen in Figure 6. The vertical tool force is set at 125 N and will be measured using a digital display torque wrench during the tool plugging stage. Non-destructive tests, macrostructural observations, and tensile strength tests were carried out to evaluate the FSW double-side welded joints. A non-destructive test (NDT) aims to ensure the absence of defects in welding specimens. This test used an X-ray generator type Eresco 65 MF4. The microstructural observation was carried out to determine changes in types, grain shape, and grain size of phases due to the double-side FSW process. The etching procedure referred to as ASTM E 407–99 Standard Practice for Metals and Alloys Microetching consists of polishing and etching using Keller's reagent, which is 5 mL $HNO_3$, 2 mL HF, 3 mL HCl, and 190 mL $H_2O$ in about 5 to 10 s [27]. The micro-Vickers hardness test was carried out with a small indenter, a load of 200 g, a holding time of 12 s, and the distance between the

test points was 2 mm. The results obtained a different distribution of hardness values in each welding area, such as base metal, heat affected zone (HAZ), thermo-mechanically affected zone (TMAZ), and weld nugget zone (WNZ), as seen in Figure 7. Tensile testing is carried out to determine the welding joint's tensile strength, yield strength, and fracture character. The specific standard for double-side friction stir welding is not yet available. So, tensile testing refers to ASTM E8 Standard Test Methods for Tension Testing [27]. Figure 8 shows this condition with the shape and dimension of the specimen.

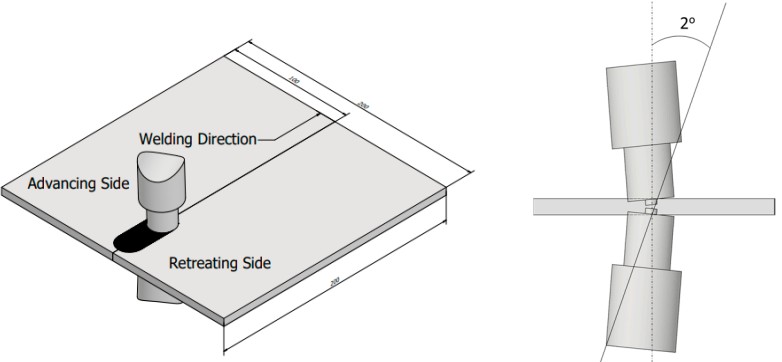

**Figure 6.** The process of the double-side FSW.

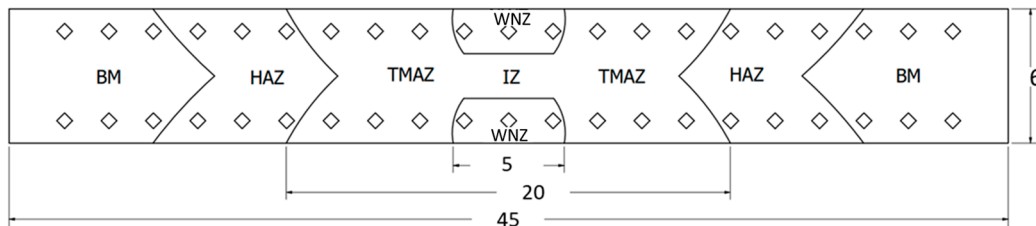

**Figure 7.** The test points of the microhardness Vickers.

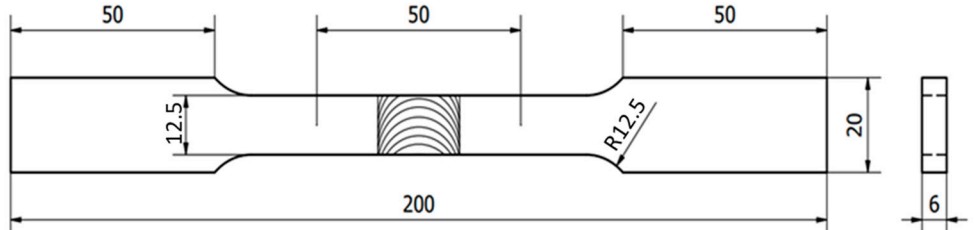

**Figure 8.** The tensile test specimen.

## 3. Apparatus Design on One-Step Double-Side FSW

Based on the problem identified, one-step double-side FSW is believed to be one of the solutions to improve FSW flexibility. Subsequently, a study has been widely performed on one-step double-side FSW with a focus on microstructure and tensile strength [28–33], material flow [34], thermal field [35], and modeling [36]. However, the study used a two-step double-side FSW method, where the first FSW welding was performed on one surface, the then welded plate was flipped over, and the second welding was performed on the other surface. The flexibility of this two-step double-side FSW method is still limited, and the FSW machine is designed to be a one-step double-side FSW, where two tools (top and bottom) simultaneously perform welding on the upper and lower surfaces. Two tools can adjust the rotational speed and direction of rotation (clockwise and counterclockwise) so the welding results can be adjusted as desired.

The standard components of the one-step double-side FSW machine include the motor, head tool, and rails. These components are then purchased, measured, and modeled in

SolidWorks software using the actual dimensions of the purchased machine components. In this modeling, non-standard components such as the table and clamping system are designed according to other components. The modeling results ensure that the movements between components can run smoothly and without colliding. The moving components in this model can be moved and viewed from various angles to check the suitability of the movement with the designed mechanism, as shown in Figure 9.

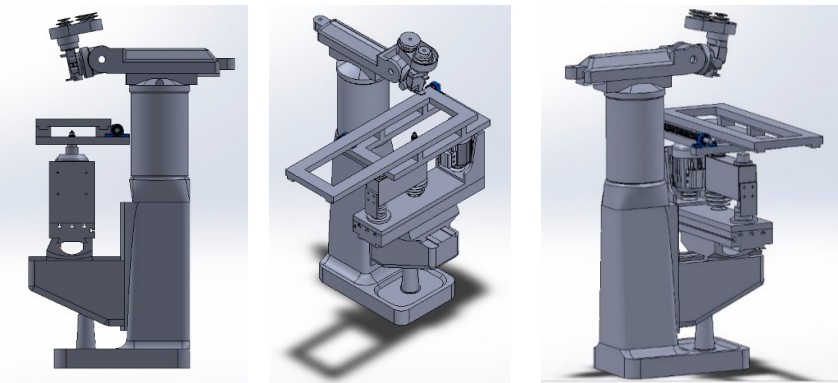

**Figure 9.** The design in different views (left, front, and right side).

After the model has been evaluated, the components of the one-step double-side FSW machine are assembled one by one to form a prototype, as seen in Figure 10. Based on the figure, the one-step double-side FSW machine has several main parts, namely the head, bed or table, and tool, with functions as follows:

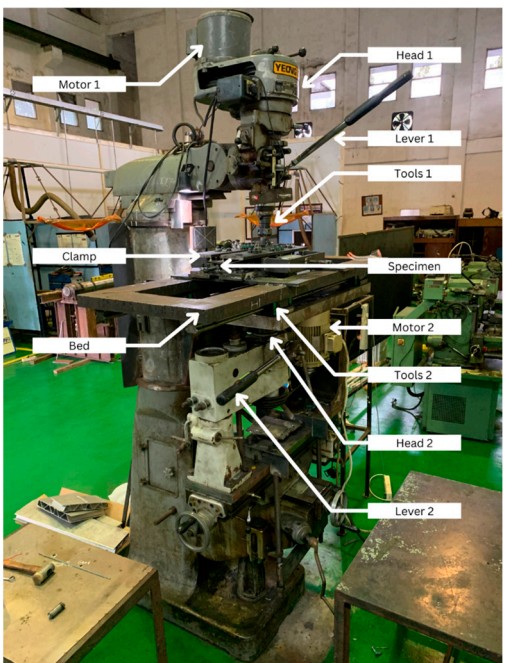

**Figure 10.** Assembled double-side FSW.

The head consists of a motor, tool post, and motion mechanism that transfers motion from the engine to the tool. Each head also has a mechanism to move tools to make contact with the surface of the workpiece. This machine has two heads called the bottom and top heads. The bottom head performs welding on the lower surface, called the 4G welding position, whereas the top head performs welding on the upper surface, called the 1G welding position. The motor power calculation of this machine is limited by several factors, including a maximum tool rotational speed of 1800 rpm, a maximum tool diameter of

20 mm, and a maximum tool pressure of 2.5 MPa. Based on the limitations, the calculations using Equations (1)–(4) obtain a compressive force on the tool of 1570 N, a friction force between the tool and the workpiece of 471 N, a torque of 4.71 Nm, and the motor power is 1.2 HP.

The bed or table is used to maintain the position of the sample during the DFSW process. This bed has a motion mechanism that moves the workpiece across the tool at a certain speed called welding speed. The clamping system also has a function to prevent the tool from hitting the workbench or other components. Figure 10 shows the specimen anvils, clamping system, and bed table, which serve to place the specimen.

The tool is used to apply friction to the sample surface and simultaneously provide stirring. Hence, the welded material flows and mixes to form a joint. To produce good material flow, the tilt angle setting needs to be adjusted. This aims to make the slope of the tools max 5°. However, in this experiment, the tilt angle of the tools is 2°. Furthermore, to perform this system, the tool is shaped so that it has two parts: the shoulder and the pin. The shoulder is the part that rubs against the sample surface to generate heat and soften the sample, whereas the pin is the protruding part on the shoulder that stir the softened material. This machine has two tools mounted on the top and bottom heads.

The one-step double-side FSW machine shown in Figure 11 is then trialed for a 6 mm AA6061 aluminum plate. Besides that, limiting the thickness aims to produce good quality welds and the machine's safety. Before the process runs, the sample is set on the table so that the top and bottom surfaces are level and aligned with the movement of the upper and lower tool. In the first trial, welding was stopped before finishing when the tool traveled approximately 20 cm because the upper tool vibrated, resulting in unwelded a disjoint on the workpiece, as shown in Figure 12.

During the FSW process, the load acting on the specimen is very complex in terms of direction and type of load, including friction force, stirring force, transverse force, tangential force, thermal force, and torque due to rotary and linear motion of the tool [37,38]. Due to the two tools worked on the specimen surfaces, the one-step double-side FSW has more complex forces than forces on the one-side FSW. These forces significantly affect the joint quality, including defects and mechanical properties of welding joints [39,40]. These forces should be directed to channel the energy produced into the workpiece joining process. The components that transfer energy, such as tools and tool posts, should be as rigid as possible to dampen transverse, lateral, and plunge forces [41,42]. Figure 13 illustrates the force applied in the one-step double-side FSW process. The traverse force acts parallel to the tool movement and is positive in the traverse direction. Since this force arises from the friction of the material to the movement of the tool, it can be expected that this force will decrease as the temperature of the material around the tool increases. An axial force is required to maintain the position of the tool on the material surface. Some friction-stir welding machines operate under load control, but in most cases, the vertical position of the tool is pre-set so that the load will vary during welding [38]. In the case of the one-step double-side FSW process, the pair of opposite forces generated by upper and lower tools are subjected to the workpiece. The torque of the rotated tool is responsible for heat generation and material flow, so torque is directly related to FSW quality. It will generate the tangential forces around the tool surface. Additionally, the torque increases with increasing motor power, decreasing tool rotational speed, and increasing the tool diameter. As the rotation tools move along the joint of the two base metals, one part is on the advancing while the other is on the retreating side.

Based on a previous study on the forces in FSW, the forces and torque should be kept within a safe range to produce a defect-free weld joint and prevent damage to the tool and apparatus. In the case of the one-step double-side FSW trial, the tool's vibration is suspected to be caused by a too-long tool post and tool. This part will be buckled when the plunging force is applied and vibrated when it is rotated, as illustrated in Figure 14.

Buckling of the beam will occur when critical force or critical beam length is exceeded. They are calculated based on the equation:

$$F = \frac{\pi^2 EI}{L^2}$$

(5)

where F is critical force, E is Young's modulus, I is moment inertia, and L is critical beam length. The critical force will be higher when the beam is shorter [43]. Unfortunately, in this study, the length of the tool post was designed to reach the specimen and allow for more space for a thick plate or a hollow extrusion panel, so it was forbidden to be cut. Therefore, the solution is not to cut the tool post but to provide flexible reinforcement so that the tool post can withstand the plunging force without buckling. The support is attached to machine bodies and tool posts, as shown in Figure 15.

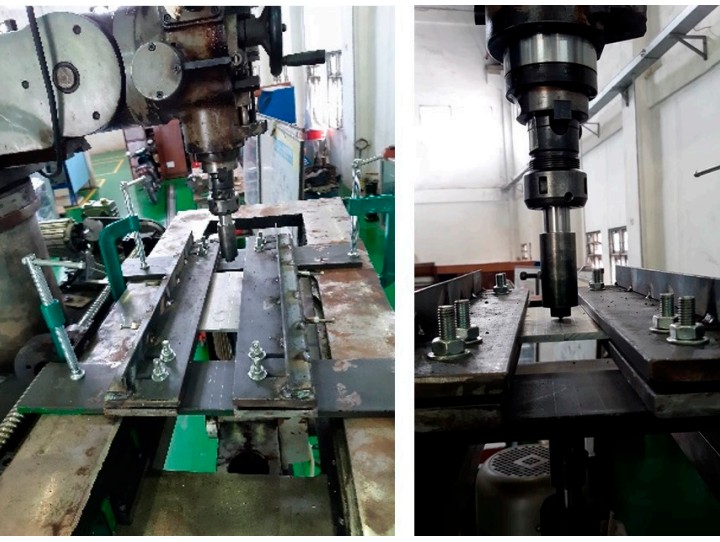

**Figure 11.** Clamping system.

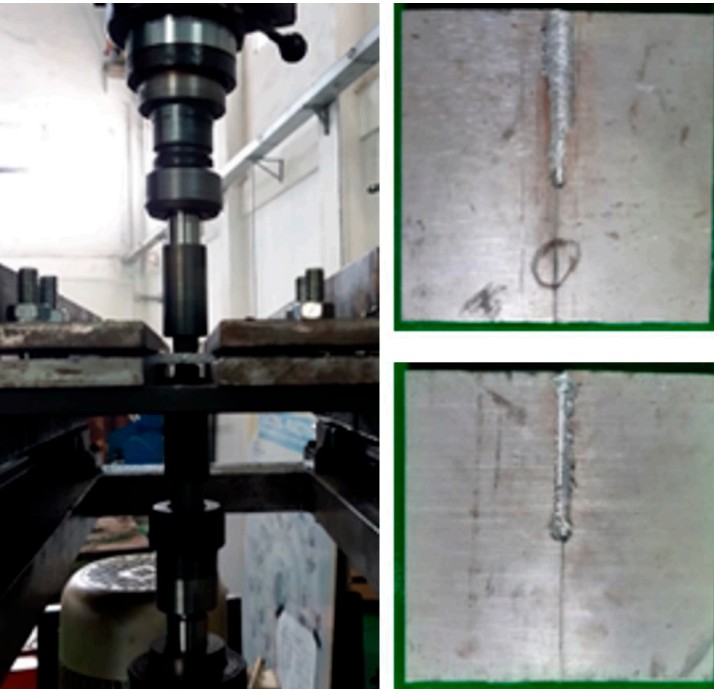

**Figure 12.** Vibration on the upper tool.

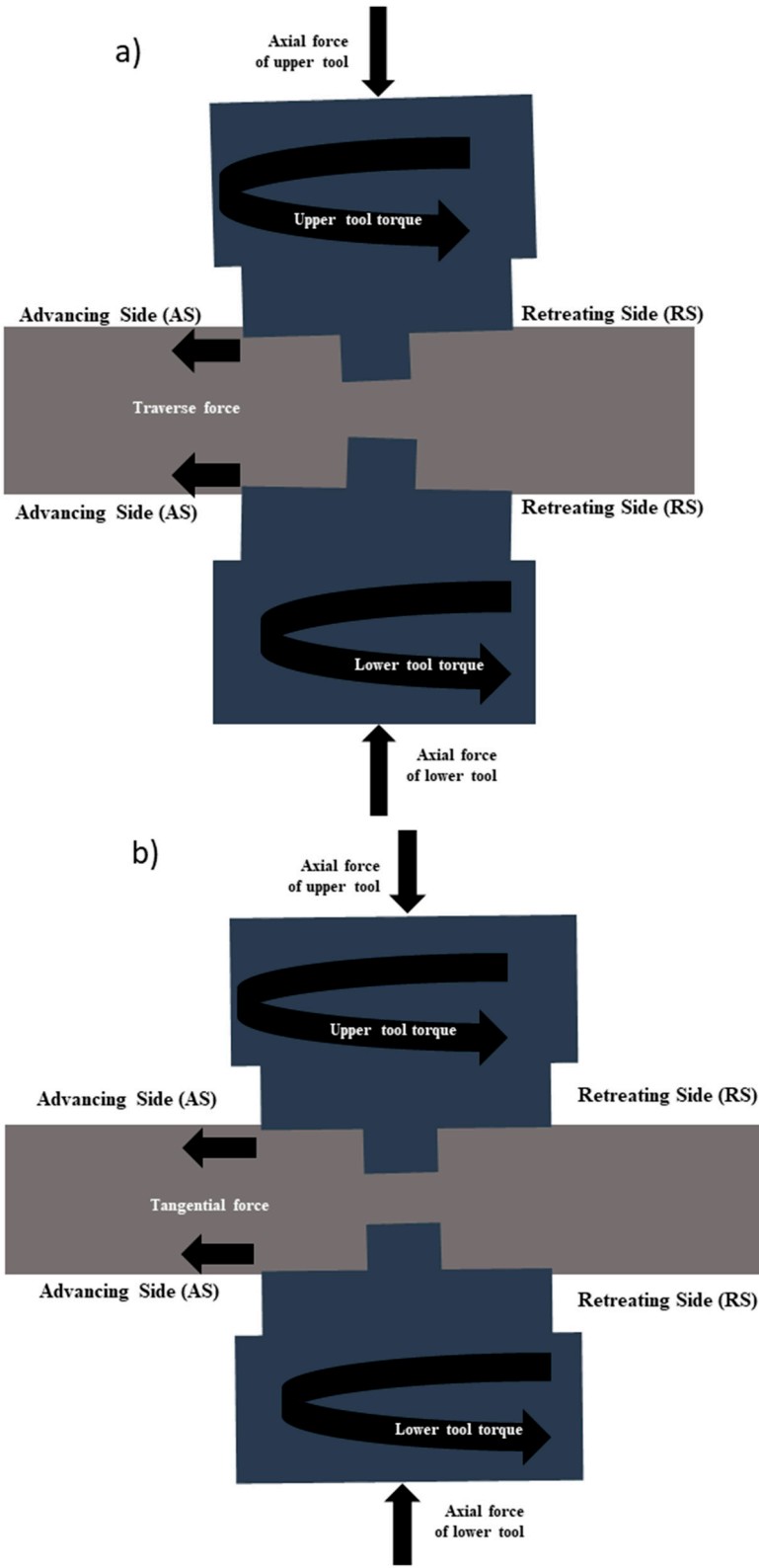

**Figure 13.** The presumed force applied in the one-step double-side FSW process: (**a**) side view, (**b**) front view.

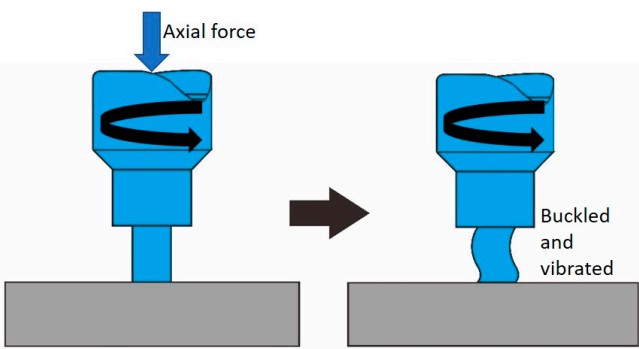

**Figure 14.** The tool experiences buckling and vibration.

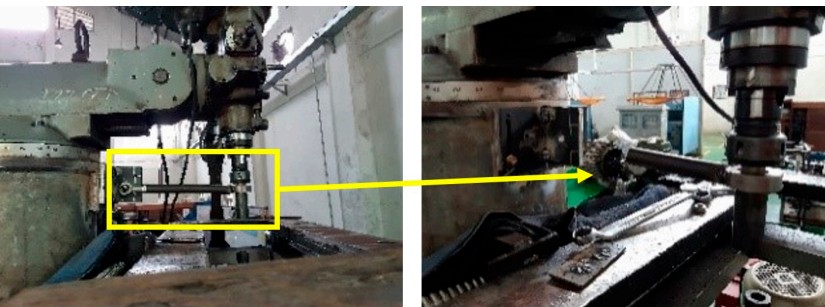

**Figure 15.** Strengthening of the upper tool.

After installing the upper tool post-reinforcement, the second welding trial is carried out with the same welding parameters as the first trial, as shown in Figure 16. The welding trial was performed in two levels of tool rotations (900/900 rpm and 1500/1500 rpm). These trials successfully joined the sample with a 40 cm long welding line on both the sides of the upper (1G position) and lower (4G position). The resulting bead is smooth, flat, and consistent in width (Figure 16). Radiographic testing, macro and microstructure observations, microhardness tests, and tensile tests were conducted to test welding results in more detail.

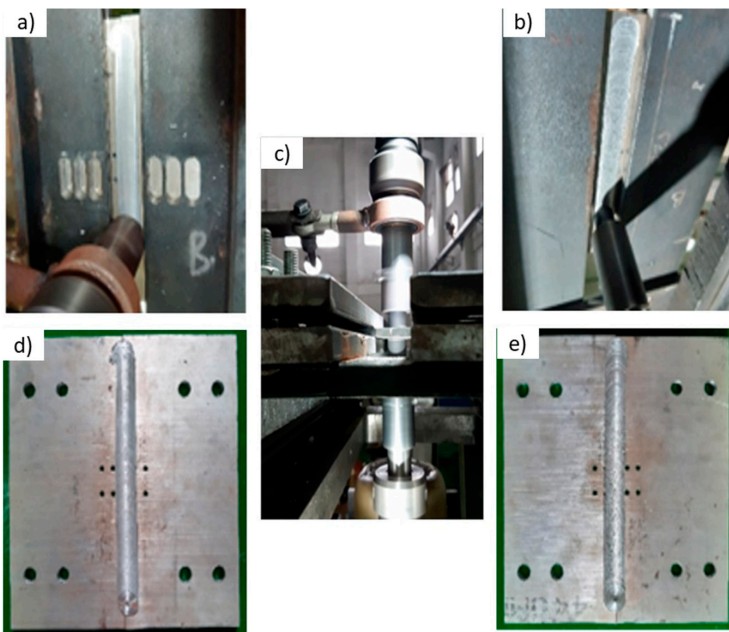

**Figure 16.** The successful joint of the one-step double-side FSW: (**a**) 1G position, (**b**) 4G position, (**c**) side view of one-step DFSW, (**d**) result 1G welding, (**e**) result 4G welding.

## 4. Characteristics of One-Step Double-Side FSW Joint

The characteristics of the welded joint in one-step double-side FSW using radiographic testing, macro and microstructure evaluation, and mechanical testing. Figure 17 shows no defects in the one-step double-side FSW joint, both with 900/900 rpm and 1500/1500 rpm, such as porosity, incomplete penetration, or incomplete fusion. Subsequently, welding aluminum with solid-state welding has been previously approved as a process with fewer porosity defects compared to fusion welding [16,31,43,44]. In solid-state welding, hydrogen gas does not dissolve in aluminum, and porosity due to trapped hydrogen gas in the molten aluminum will not occur [45,46]. This is reinforced by the results of macrostructure observations, as shown in Figure 18. It shows the cross-sectional macrostructure of the one-step double-side FSW joint. Several zones are formed, namely the nugget zone (NZ), the thermo-mechanically affected zone (TMAZ), and the heat-affected zone (HAZ). Due to two tools working in the workpiece, NZ consists of upper NZ and lower NZ.

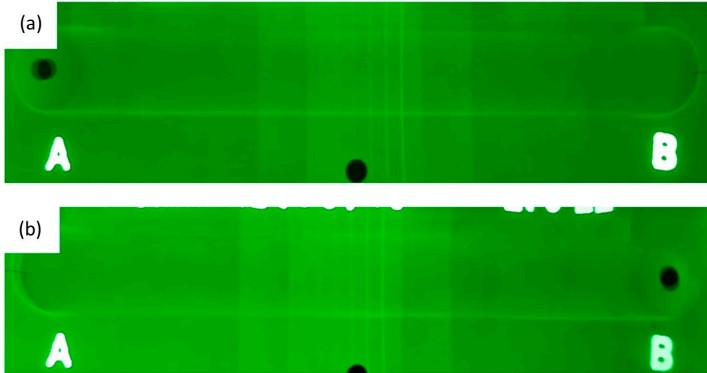

**Figure 17.** Result of radiography test on the one-step double-side FSW joint: (**a**) tool rotation speed of 900/900 rpm, (**b**) tool rotation speed of 1500/1500 rpm.

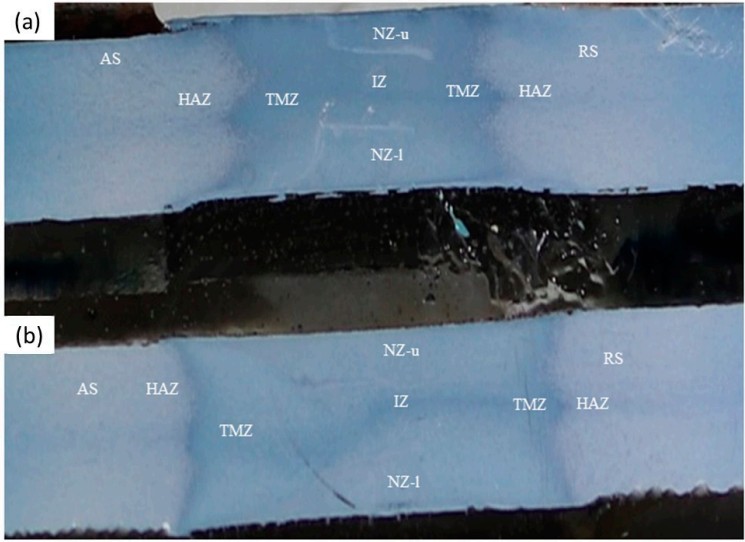

**Figure 18.** Macrostructure of the one-step double-side FSW joint: (**a**) tool rotation speed of 900/900 rpm, (**b**) tool rotation speed of 1500/1500 rpm.

Furthermore, there is a new zone on the one-step double-side FSW joint located between the upper and lower NZ, named the interaction zone (IZ). The IZ does not experience material mixing but is exposed to heat and mechanical force from the two rotating pins. Figure 18 also shows the higher the tool rotation speed causes the wider HAZ and TMAZ. The heat input increases as increasing tool rotation speed and widens HAZ and TMAZ [47].

The microstructure of the base metal AA6061 can be seen in Figure 19. Grains tend to be flatly elongated due to the influence of the cold rolling process during manufacturing. Microstructure consists of two distinct phases: aluminum alloy matrix ($\alpha$-Al) and precipitate ($\beta''$-Mg5Si6), which are well dispersed in the matrix. The precipitation phase occurs as a result of the T6 heat treatment of the aluminum alloy [48].

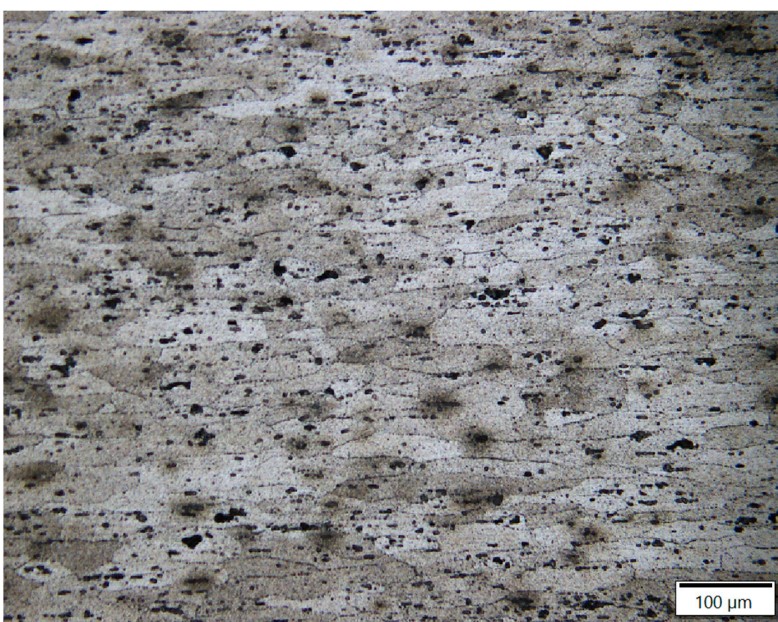

**Figure 19.** Microstructure of the base metal AA6061.

Microstructures of each zone for both tool rotation speed of 900/900 rpm and 1500/1500 rpm is summarized in Table 2. The NZ is subjected to plastic deformation, thermal cycling, and material movement due to friction and stir from shoulders and pins. This zone undergoes significant dynamic recrystallization resulting in a fine-grain structure [49]. The precipitate lead to the evolution of complex microstructures, including the dissolution of the precipitate from artificial aging processes [17]. In general, the grain size formed in NZ will be larger as the heat input increases due to the tool rotation speed increasing. There is no significant difference in grain size between the upper (NZ-u) and lower (NZ-l). TMAZ experiences plastic deformation and thermal cycling due to friction and mechanical force from the shoulder but does not experience material movement due to stirring, so it does not undergo dynamic recrystallization [49]. However, the re-dissolving of the precipitate occurred due to high-temperature exposure during FSW. The amount of solubility depends on the thermal cycle in TMAZ [24]. In general, the grain size of TMAZ will increase with increasing heat input caused by increasing the tool rotation speed. HAZ undergoes limited plastic deformation and recrystallization so that it has almost the same grain as the base metal but is coarser [25]. In general, HAZ changes phase grains due to the influence of heat from welding, where the grains that are flatly elongated due to the cold rolling of the base metal will return to the shape before the rolling process. The IZ consists of grains that are plastically deformed by the heat and mechanical forces of the pins. IZ does not undergo recrystallization, so it has a larger grain size than NZ grains. IZ grain size differs greatly from the thermo-mechanically affected zone (TMAZ).

**Table 2.** Microstructure of the zones in one-step double-side FSW joint.

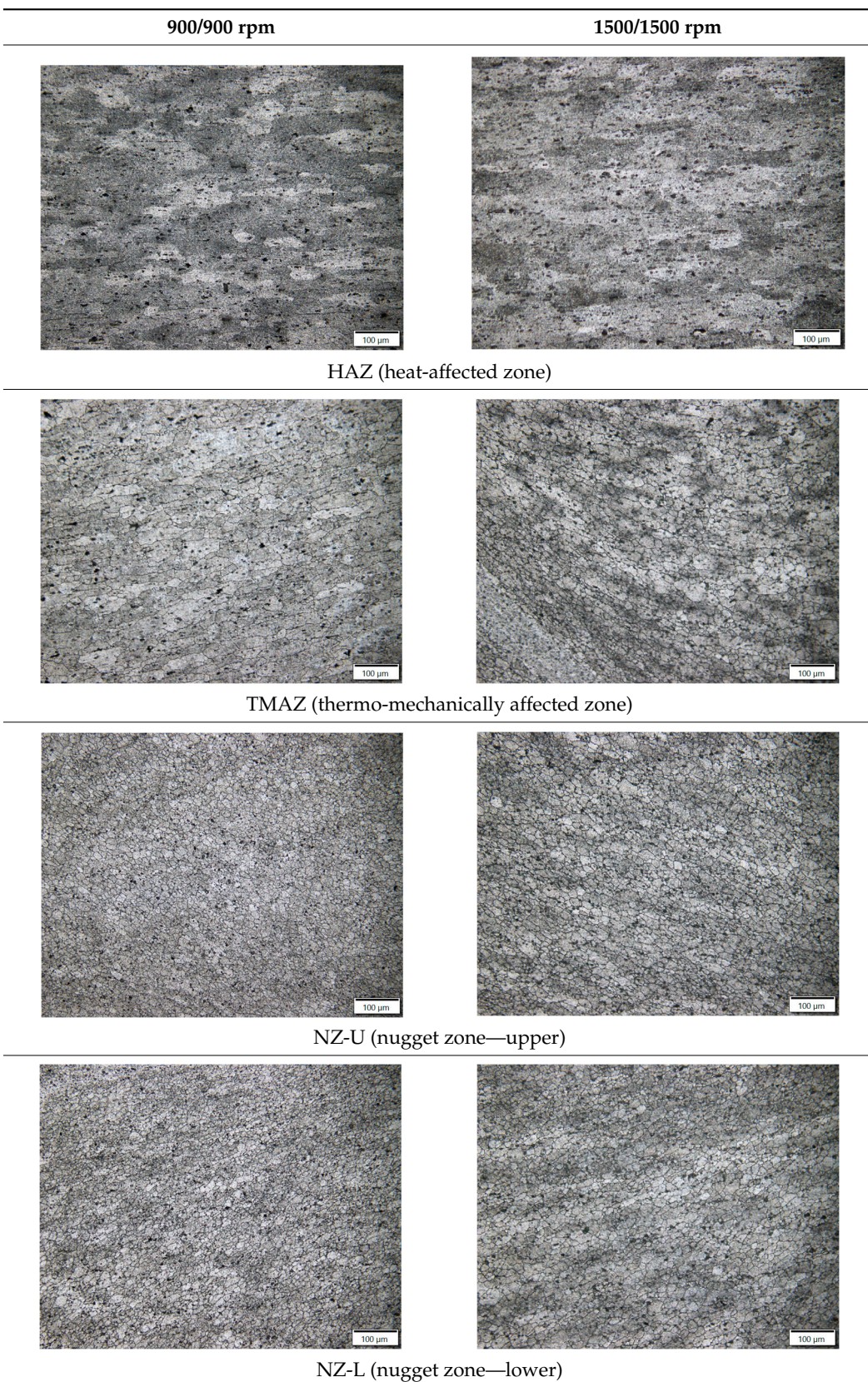

| 900/900 rpm | 1500/1500 rpm |
|:-:|:-:|
| HAZ (heat-affected zone) ||
| TMAZ (thermo-mechanically affected zone) ||
| NZ-U (nugget zone—upper) ||
| NZ-L (nugget zone—lower) ||

**Table 2.** *Cont.*

| 900/900 rpm | 1500/1500 rpm |
|:---:|:---:|
| 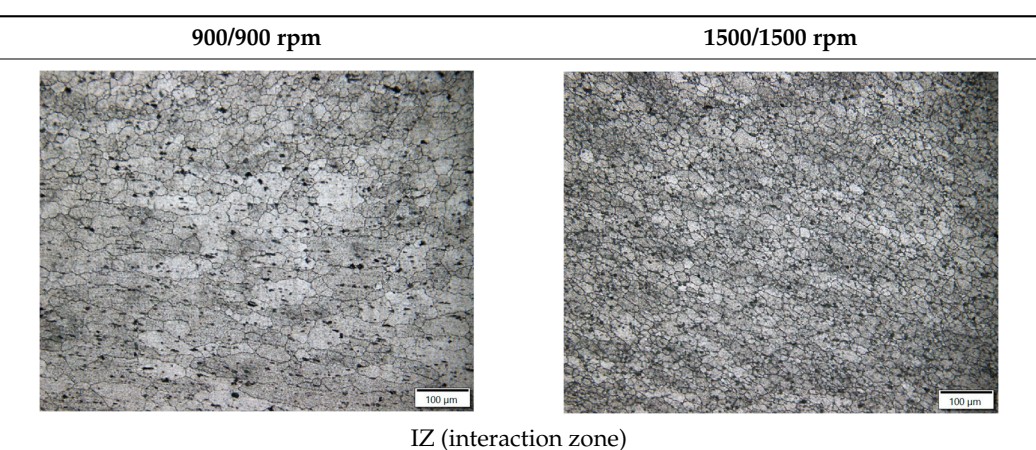 | |
| IZ (interaction zone) | |

Figure 20 shows the hardness distribution of the upper and lower parts of one-step double-side FSW. From this figure, there is no difference in the hardness of the upper and lower part. The hardness decreases in the TMAZ area and slowly increases at the transition between the HAZ and TMAZ. This decrease in hardness is due to heat, plastic deformation, and recrystallization that occur during welding, eliminating the effects of cold working and precipitation hardening beforehand [50]. The NZ hardness sharply increases in the middle, forming a hardness distribution in a 'W' profile. This hardness profile usually occurs in heat-treatable aluminum alloys. The average NZ hardness of the FSW joint with a tool rotational speed of 1500/1500 rpm is around 63 VHN. Due to the larger grain size, it is softer than an FSW joint with a tool rotational speed of 900/900 rpm. Both specimens have lower hardness when compared to the base metal hardness.

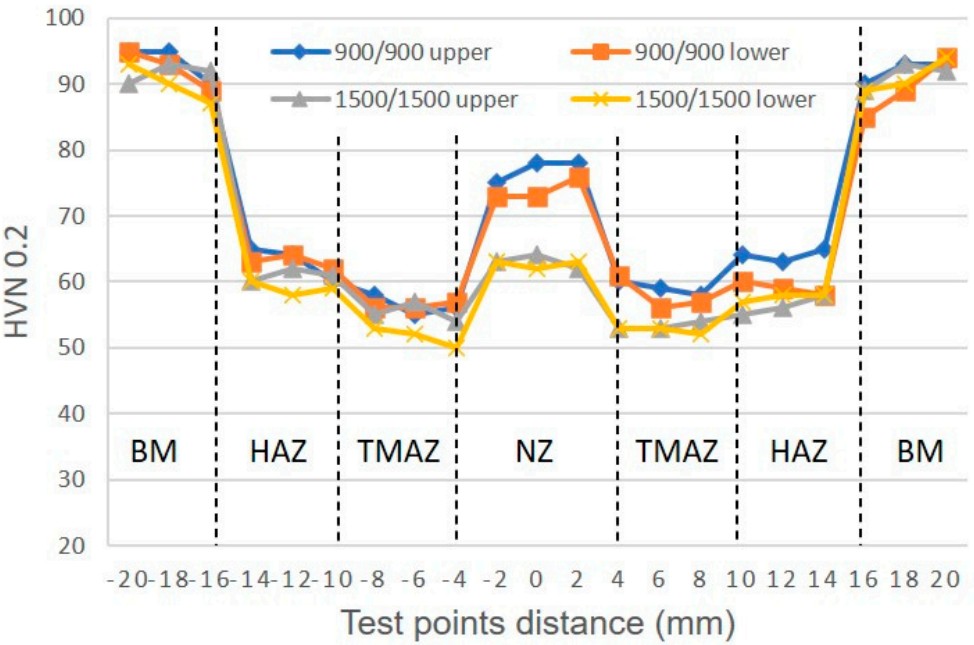

**Figure 20.** Hardness distribution of the zones in one-step double-side FSW joint.

Figure 21 shows the average maximum tensile strength of the base metal and one-step double-side FSW joints. The tensile strength of the one-step double-side FSW joints for all variations of tool rotation speed is lower than that of the base metal. The joint efficiency is defined as comparing the tensile strength of the weld joint to the base metal. The tool rotation speed of 1500 rpm produces better joint efficiency than 900 rpm. When

the tool rotation speed is too low, such as 900 rpm, the material mixing becomes uneven and relatively less heat input during welding, resulting in incomplete recrystallization and a decrease in tensile strength. The hardness test results show that tensile strength is reduced due to softening zones appearing. The degree of softening increases and is directly proportional to the increasing tool rotational speed [45]. When AA 6061-T6 is exposed to temperatures above 200 °C, for example, during welding, dissolution of the reinforcing deposits occurs (i.e., loss of the T6 state). For example, FSW of aluminum alloys occurs at temperatures around 400–450 °C, which will make the dissolution of the reinforcement precipitate and cause the mechanical properties of joints to decrease. Despite significant grain refinement, the loss of tensile strength indicates that the precipitate reinforcement phase is the main strengthening mechanism in AA 6061-T6 [49]. The effect of this precipitate mechanism causes an increase in the mechanical properties of the weld zone [50].

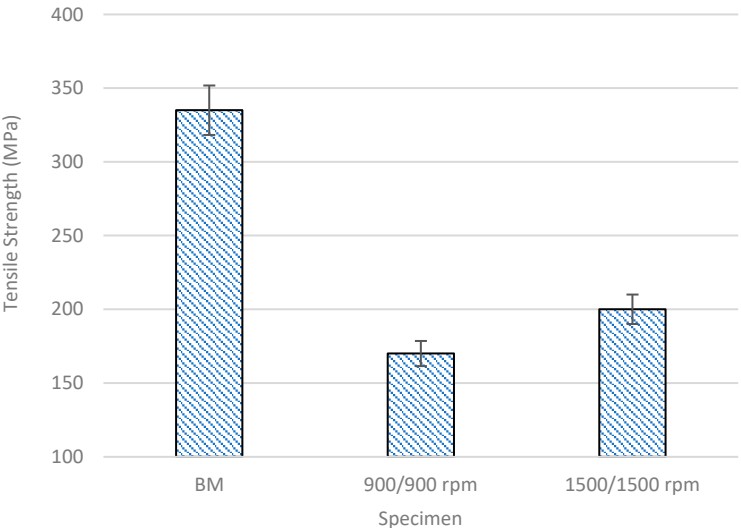

**Figure 21.** Tensile strength of one-step double-side FSW joint.

Figure 22 shows that all specimens were brittle and fractured. The brittle fracture structure consists of a predominance of the cleavage structure and only a slight dimple. It indicates that the specimen is low fracture toughness, hard but brittle [51]. It occurs by direct crystallographic separation due to the loss of ionic bonds and makes a trans-granular fracture, where the fracture cuts through the grain. It is characterized by multifaceted fracture texture due to the orientation difference between grain and grain.

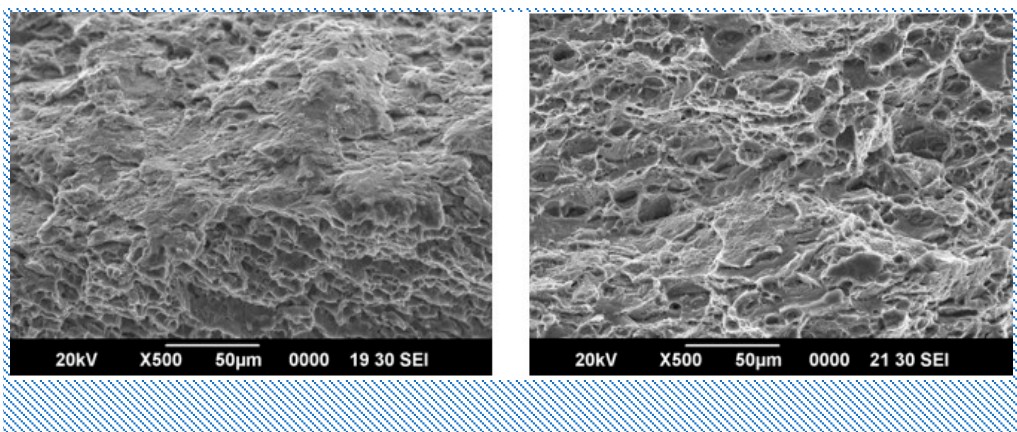

**Figure 22.** The fracture surface of tensile test: (**a**) tool rotation speed of 900/900 rpm, (**b**) tool rotation speed of 1500/1500 rpm.

## 5. Conclusions

A one-step double-side FSW apparatus was designed and made, and through welding trials using various parameters, successful welding of 6 mm aluminum was achieved without any porosity defects. The specifications of this apparatus are a maximum tool rotation speed of 1800 rpm, maximum tool diameter of 20 mm, and pressure of tool of 2.5 MPa. The clamping system has proven to influence the welding process and its results. Due to the complexity of the load and the direction of movement during welding, strong vibrations were generated, which needed to be addressed by the clamping system. As a result, future work should focus on designing a strong and rigid clamping system that is also easy to install and remove, enhancing the flexibility of the one-step double-side FSW process. The results of one-step double-side FSW with different tool rotational speeds show that the degree of softening increases and is directly proportional to the increase in tool rotational speed. It caused the resulting tensile strength value to be higher. In addition, the hard test resulted that one-step double-side FSW having a lower hardness value when compared to the hardness of the base metal. Therefore, the one-step double-side FSW method was the best solution to produce aluminum welded joints with minimal defects and higher tensile strength.

**Author Contributions:** Conceptualization, T. and J.; methodology, N.M.; software, A.R.P.; validation, P.P.; investigation, E.D.W.S.P., Y.P.D.S.D. and H.; writing—original draft preparation, T.; writing—review and editing, T. All authors have read and agreed to the published version of the manuscript.

**Funding:** The apparatus was funded by BRIN through RIIM schema, grant number: 1562.1/UN27.22/HK.07.00/2022, while Universitas Sebelas Maret funded research activities through Indonesian Collaboration Research (RKI) 2022, grant number: 872.1/UN27.22/PT.01.03/2022.

**Institutional Review Board Statement:** Not applicable.

**Informed Consent Statement:** Not applicable.

**Data Availability Statement:** Not available.

**Acknowledgments:** The authors would like to thank Universitas Sebelas Maret Surakarta and Badan Riset dan Inovasi Nasional, Indonesia, for providing many facilities and financially supporting through RKI 2022 (Contract No. 872.1/UN27.22/PT.01.03/2022) and RIIM 2022 grant (Contract No. 1562.1/UN27.22/HK.07.00/2022), Isworo Jati for preparing equipment set-ups, and all students in the METALS research group for assisting this research.

**Conflicts of Interest:** The authors declare no conflict of interest.

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
