# Peer review of "Apparatus Design of One-Step Double-Side Friction Stir Welding for Aluminum Plates"

_designs_

Round 1
Reviewer 1 Report
As part of this scientific study, a double acting FSW was carried out and assembled, and welding tests were carried out in various welding modes. Interesting results are obtained, but there are a number of questions:
1. I propose to add scale marks in fig. 1.
2. For all decimal fractions used in the text, I propose to use not "," but "."
3. What are the standards for double side friction stir welding and mechanical testing was used ?
4. Are the obtained mechanical properties of the weld zone sufficient?
5. What is the scientific novelty of the obtained results? What is better than existing technologies was obtained?
6. The abstract and conclusions of the article should contain information about new results and methods for obtaining them. They cannot be a description of existing problems, but should show their solution.
Author Response
Dear Reviewer
thanks for the valuable comments. We have responded to your comments one by one and we submit them in a separate file. So please see the attachment. In response to your comment, we mark color letters in the attached text.
Thank you
Best regards
Triyono

Reviewer 2 Report
Dear Authors
Friction welding technology and the machines used in this technology are well known and described in the literature. However, the paper submitted for review contains a new concept to improve the quality and refinement of this technology. However, the work requires a lot of adjustments. The most important issue is the division of Chapter 3. Results into two parts, i.e. two separate chapters. The first part would contain content regarding the design and construction of the test stand, i.e. the described device, and the second part would contain only a description of the test results. So in a separate chapter design and in a separate chapter test results. Detailed comments are provided below.
1. Line 93-94 - it was written "Furthermore, a stress analysis was performed on the main parts…", but nowhere in the article is the stress analysis performed; therefore it should be supplemented.
2. Fig. 3. - the font of the procedure descriptions should be increased. You can increase the width of the drawing to make it more readable (visible).
3. Line 113 - Please justify in the article why this alloy was selected for testing, not another.
4. Line 114 and 120 - please cite the literature on the basis of which the percentage composition of the material was given, e.g. the manufacturer's datasheet.
5. Lines 129 and 137 - please put the title of the standard in references at the end of the article.
6. Fig. 5 and 6 - please increase the font of the sketch dimensions - it is hardly visible.
7. Fig. 6 - please correct the inscription in the drawing - it should be written as WNZ instead of NWZ; the drawing description should be consistent with the description in the text (Line 135).
8. Fig. 7. - please change the "comma" to "dot" in the notation of the dimension "12.5".
9. Figures 8 and 9 are unnecessary as they add nothing to the article.
10. Fig. 14. - please increase the font in the description of the drawing - it is hardly visible, and it is difficult to read.
11. Lines 239 and 240 - the description of the individual variables in formula (1) should be written below the formula, as is commonly used.
12. Table 2. Photographs of the structures are of poor quality and hardly visible. This should be corrected, e.g. enlarge the photos.
13. Fig. 21. - Were tests performed on only one sample? If there were more samples, then the following question is - what is the average microhardness in each zone and what was the standard deviation?
14. Fig. 22. Tensile test results are very poorly documented. The article does not specify the number of samples. The graph refers only to Rm, and where are the other parameters? It is best to show tension diagrams, then we have a picture of stresses and strains.
15. Conclusions on test results should be completed. Only in the last sentence is the higher tensile strength briefly mentioned. So why was the study carried out if no reference was made to the results in the conclusions?
Author Response
Dear Reviewer
Please see the attachment
Thank you
Best regards
Triyono

Reviewer 3 Report
Paper No.: designs-2374743
Title: Apparatus Design of Double Side Friction Stir Welding for Thick Plate Aluminum
The subject is good, but the presentation needs to be improved and more details should be presented about the design and basis of motors selection and materials. The attained power of the machine, and the limits of application should be described. Some of the noted comments are mentioned below.
Abstract
1- The word “carbody” is two words “car body”.
2- Please specify, where in the car body can aluminum alloys be applied?
3- Sheet thickness of 6mm is not thick and not representing challenge in FSW, so that I suggest not to stress on “thick plate) and modify the title to: Apparatus Design of Double Side Friction Stir Welding for Aluminum Plate.
4- Later, this apparatus can be tested on FSW of plates 10 mm or thicker.
Introduction
5- Please use reference for Figure 1
6- Figure 2.b in not well representing the proposed FSW method, especially regarding the how to power the welding direction (feeding).
7- Consuming two thirds of the introduction about the advantage and disadvantage of the well-established FSW is not good. Please introduce more about the development in the FSW machines, not FSSW machine.
Materials and Methods
Apparatus Design and Assembling
8- In Figure 3, the left side and middle columns can be merged in one column.
9- Use higher font size for the right-side column to be visible and comparable with the manuscript font size.
10- “The selection of the material was also carried out based on the modeling results”. No thing clear about which materials selected? and why?
11- The planned torque and power of the machine, and the specification of the used motors are not mentioned.
12- The application limits of the machines regarding welded materials, and plate thickness also not specified.
13- Can the bottom and top head be tilted? and how much the degree of tilting?
14- The power (motor) controlling the bed is not clear.
Results
15- Could the motors rotation direction be reversed?
16- Text on Figure 14 is very small and not clear.
17- It was enough to judge the designed apparatus to present visual appearance, macro-optical cross sections of the welding nugget, and the mechanical (tensile) testing. Microstructure and fracture investigation in not well related to the study and not supporting idea of machine design.
Conclusions:
Furthermore, the higher tool rotation speed of FSW, the coarser grain size and the higher tensile strength. How it comes? Coarser grain size results in lower strength
Author Response

(The authors gave the same response as above.)

Round 2
Reviewer 1 Report
The authors took into account all my recommendations. I also suggest showing in fig. 21 fracture surface at the macro- and micro- levels. As known from the article: https://www.mdpi.com/1996-1944/12/13/2051, this will allow a deeper understanding of the micromechanisms of material fracture.
Author Response
Dear reviewer
Thank you for recommendation. We have added the macro and micro level of fracture surface (page 16 line 386). We also have cited the paper you have sugested.
Than you
Best reagards
Triyono

Reviewer 2 Report
Dear Authors,
After the corrections are made, I will recommend the article for publication.
Kind regards
Reviewer
Author Response
Dear reviewer
Thank you for your decision
Best regards
Triyono
Reviewer 3 Report
Paper No.: designs-2374743
Title: Apparatus Design of Double Side Friction Stir Welding for Thick Plate Aluminum
Many points either totally not responded or not well treated regarding the main line of the article namely the machine design:
- basis of motors selection and machine parts material,
- the attained power of the machine, and
- the limits of application should be described.
- Some of the noted comments were not treated at all and others were partially treated.
- The title is not modified yet
- Side-wall, and roof of the cars are usually manufactured from steel sheet not from aluminum plates.
- In your response: “The selection of the material was also carried out based on the stress analysis from modeling results. It aims to reduce stress concentrations, deflections, thermal gradients, and potential failure points in the machine. This process can help identify any unforeseen challenges or limitations before committing to full-scale production”. No evidence provided in the article for carrying out such modelling or simulation.
- The planned torque and power of the machine, and the specification of the used motors are not mentioned. Your response should be implemented in the article and supported by equations and their results. How the value 2 Hp is predicted?
Author Response
Dear reviewer
Thank you for your important comments
Please see the attachment.
Thank you
Best regards
Triyono

Round 3
Reviewer 3 Report
accepted